# Sequential Coordination of Deep Models for Learning Visual Arithmetic

## Abstract

Achieving machine intelligence requires a smooth integration of perception and reasoning, yet models developed to date tend to specialize in one or the other; sophisticated manipulation of symbols acquired from rich perceptual spaces has so far proved elusive. Consider a visual arithmetic task, where the goal is to carry out simple arithmetical algorithms on digits presented under natural conditions (e.g. hand-written, placed randomly). We propose a two-tiered architecture for tackling this problem. The lower tier consists of a heterogeneous collection of information processing modules, which can include pre-trained deep neural networks for locating and extracting characters from the image, as well as modules performing symbolic transformations on the representations extracted by perception. The higher tier consists of a controller, trained using reinforcement learning, which coordinates the modules in order to solve the high-level task. For instance, the controller may learn in what contexts to execute the perceptual networks and what symbolic transformations to apply to their outputs. The resulting model is able to solve a variety of tasks in the visual arithmetic domain, and has several advantages over standard, architecturally homogeneous feedforward networks including improved sample efficiency.

## 1 Introduction

Recent successes in machine learning have shown that difficult perceptual tasks can be tackled efficiently using deep neural networks (LeCun et al., 2015). However, many challenging tasks may be most naturally solved by combining perception with symbol manipulation. The act of grading a question on a mathematics exam, for instance, requires both sophisticated perception (identifying discrete symbols rendered in many different writing styles) and complex symbol manipulation (confirming that the rendered symbols correspond to a correct answer to the given question). In this work, we address the question of creating machine learning systems that can be trained to solve such perceptuo-symbolic problems from a small number of examples. In particular, we consider, as a first step toward full-blown exam question grading, the *visual arithmetic* task, where the goal is to carry out basic arithmetic algorithms on hand-written digits embedded in an image, with the wrinkle that an additional symbol in the image specifies which of a handful of algorithms (e.g. max, min, +, *) should be performed on the provided digits.

One straightforward approach to solving the visual arithmetic task with machine learning would be to formulate it as a simple classification problem, with the image as input, and an integer giving the correct answer to the arithmetic problem posed by the image as the label. A convolutional neural network (CNN; LeCun et al., 1998) could then be trained via stochastic gradient descent to map from input images to correct answers. However, it is clear that there is a great deal of structure in the problem which is not being harnessed by this simple approach, and which would likely improve the sample efficiency of any learning algorithm that was able to exploit it. While the universal approximation theorem (Hornik et al., 1989) suggests that an architecturally homogeneous network such as a CNN should be able to solve any task when it is made large enough and given sufficient data, imposing model structure becomes important when one is aiming to capture human-like abilities of strong generalization and learning from small datasets (Lake et al., 2016).

In particular, in this instance we would like to provide the learner with access to modules implementing information processing functions that are relevant for the task at hand — for example, modules that classify individual symbols in the image, or modules that perform symbolic computations on stored representations. However, it is not immediately clear how to include such modules in standard deep networks; the classifiers need to somehow be applied to the correct portion of the image, while the symbolic transformations need to be applied to the correct representations at the appropriate time and, moreover, will typically be non-differentiable, precluding the possibility of training via backpropogation.

In this work we propose an approach that solves this type of task in two steps. First, the machine learning practitioner identifies a collection of modules, each performing an elementary information processing function that is predicted to be useful in the domain of interest, and assembles them into a designed information processing machine called an *interface* (Zaremba et al., 2016) that is coupled to the external environment. Second, reinforcement learning (RL) is used to train a controller to make use of the interface; use of RL alleviates any need for the interface to be differentiable. For example, in this paper we make use of an interface for the visual arithmetic domain that contains: a discrete attention mechanism; three pre-trained perceptual neural networks that classify digits/classify arithmetic symbols/detect salient locations (respectively); several modules performing basic arithmetic operations on stored internal representations. Through the use of RL, a controller learns to sequentially combine these components to solve visual arithmetic tasks.

**Contributions.** *We propose a novel recipe for constructing agents capable of solving complex tasks by sequentially combining provided information processing modules.* The role of the system designer is limited to choosing a pool of modules and gathering training data in the form of input-output examples for the target task. A controller is then trained by RL to use the provided modules to solve tasks. We evaluate our approach on a family of visual arithmetic tasks wherein the agent is required to perform arithmetical reduction operations on hand-written digits in an image. Our experiments show that the proposed model can learn to solve tasks in this domain using significantly fewer training examples than unstructured feedforward networks.

The remainder of the article is organized as follows. In Section 2 we describe our general approach and lay down the required technical machinery. In Section 3 we describe the visual arithmetic task domain in detail, and show how our approach may be applied there. In Section 4 we present empirical results demonstrating the advantages of our approach as it applies to visual arithmetic, before reviewing related work in Section 5 and concluding with a discussion in Section 6.

## 2 TECHNICAL APPROACH

### 2.1 PARTIALLY OBSERVABLE MARKOV DECISION PROCESSES

Our approach makes use of standard reinforcement learning formalisms (Sutton and Barto, 1998). The external world is modelled as a Partially Observable Markov Decision Process (POMDP), $\mathcal{E}$. Each time step $\mathcal{E}$ is in a state $s_t$, based upon which it emits an observation $o_t$ that is sent to the learning agent. The agent responds with an action $a_t$, which causes the environment to emit a reward $r_t$. Finally, the state is stochastically updated according to $\mathcal{E}$'s dynamics, $s_{t+1} \sim P(\cdot|s_t, a_t)$. This process repeats for $T$ time steps. The agent is assumed to choose $a_t$ according to a parameterized policy that maps from observation-action histories to distributions over actions, i.e. $a_t \sim \pi_\theta(\cdot|h_t)$, where $h_t = o_0, a_0, \ldots, o_t$ and $\theta$ is a parameter vector.

### 2.2 INTERFACES

We make extensive use of the idea of an *interface* as proposed in Zaremba et al. (2016). An interface is a designed, domain-specific machine that mediates a learning agent's interaction with the external world, providing a representation (observation and action spaces) which is intended to be more conducive to learning than the raw representation provided by the

external world. In this work we formalize an interface as a POMDP $\mathcal{I}$ distinct from $\mathcal{E}$, with its own state, observation and action spaces. The interface is assumed to be coupled to the external world in a particular way; each time step $\mathcal{E}$ sends an observation to $\mathcal{I}$, which potentially alters its state, after which $\mathcal{I}$ emits its own observation to the agent. When the agent responds with an action, it is first processed by $\mathcal{I}$, which once again has the opportunity to change its state, after which $\mathcal{I}$ sends an action to $\mathcal{E}$. The agent may thus be regarded as interacting with a POMDP $\mathcal{C}$ comprised of the combination of $\mathcal{E}$ and $\mathcal{I}$. $\mathcal{C}$'s observation and action spaces are the same as those of $\mathcal{I}$, its state is the concatenation of the states of $\mathcal{I}$ and $\mathcal{E}$, and its dynamics are determined by the nature of the coupling between $\mathcal{I}$ and $\mathcal{E}$.

Zaremba et al. (2016) learn to control interfaces in order to solve purely algorithmic tasks, such as copying lists of abstractly (rather than perceptually) represented digits. One of the main insights of the current work is that the idea of interfaces can be extended to tasks with rich perceptual domains by incorporating pre-trained deep networks to handle the perceptual components.

## 2.3 ACTOR-CRITIC REINFORCEMENT LEARNING

We train controllers using the actor-critic algorithm (see e.g. (Sutton and Barto, 1998; Degris et al., 2012; Mnih et al., 2016; Schulman et al., 2015)). We model the controller as a policy $\pi_\theta$ that is differentiable with respect to its parameters $\theta$. Assume from the outset that the goal is to maximize the expected sum of discounted rewards when following $\pi_\theta$:

$$J(\theta) = E_{\tau \sim \pi_\theta} \left[ \sum_{t=0}^{T-1} \gamma^t r_t \right] = \sum_{\tau} P_{\pi_\theta}(\tau) \sum_{t=0}^{T-1} \gamma^t r_t \tag{1}$$

where $\tau = (o_0, a_0, r_0, \ldots, o_{T-1}, a_{T-1}, r_{T-1})$ is a trajectory, $P_{\pi_\theta}(\tau)$ is the probability of that trajectory under $\pi_\theta$, and $\gamma \in (0, 1]$ is a discount factor. We look to maximize this objective using gradient ascent; however, it is not immediately clear how to compute $\nabla_\theta J(\theta)$, since the probability of a trajectory $P_{\pi_\theta}(\tau)$ is a function of the environment dynamics, which are generally unknown. Fortunately, it can be shown that an unbiased estimate of $\nabla_\theta J(\theta)$ can be obtained by differentiating a surrogate objective function that can be estimated from sample trajectories. Letting $R_t = \sum_{i=t}^{T-1} \gamma^{i-t} r_i$, the surrogate objective is:

$$\mathcal{F}(\theta) = E_{\tau \sim \pi_\theta} \left[ \sum_{t=0}^{T-1} \log(\pi_\theta(a_t | h_t)) R_t \right] \tag{2}$$

The standard REINFORCE algorithm (Williams, 1992) consists in first sampling a batch of trajectories using $\pi_\theta$, then forming an empirical estimate $f(\theta)$ of $\mathcal{F}(\theta)$. $\nabla_\theta f(\theta)$ is then computed, and the parameter vector $\theta$ updated using standard gradient ascent or one of its more sophisticated counterparts (e.g. ADAM; Kingma and Ba (2014)).

The above gradient estimate is unbiased (i.e. $E[\nabla_\theta f(\theta)] = \nabla_\theta J(\theta)$) but can suffer from high variance. This variance can be somewhat reduced by the introduction of a baseline function $b_t(h)$ into Equation (2):

$$\mathcal{F}(\theta) = E_{\tau \sim \pi_\theta} \left[ \sum_{t=0}^{T-1} \log(\pi_\theta(a_t | h_t)) \left( R_t - b_t(h_t) \right) \right] \tag{3}$$

It can be shown that including $b_t(h)$ does not bias the gradient estimate and may lower its variance if chosen appropriately. The value function for the current policy, $V^{\pi_\theta}(h) = E_{\tau \sim \pi_\theta}[R_t | h_t = h]$, is a good choice for $b_t$ (Sutton et al., 2000) — however this will rarely be known. A typical compromise is to train a function $V_\omega(h)$, parameterized by a vector $\omega$, to approximate $V^{\pi_\theta}(h)$ at the same time as we are training $\pi_\theta$. Specifically, this is achieved by minimizing a sample-based estimate of $E_{\tau \sim \pi_\theta, t} \left[ (R_t - V_\omega(h_t))^2 \right]$.

We employ two additional standard techniques (Mnih et al., 2016). First, we have $V_\omega(h)$ share the majority of its parameters with $\pi_\theta$, i.e. $\omega = \theta$. This allows the controller to learn

useful representations even in the absence of reward, which can speed up learning when reward is sparse. Second, we include in the objective a term which encourages $\pi_\theta$ to have high entropy, thereby favouring exploration.

Overall, given a batch of $N$ sample trajectories from policy $\pi_\theta$, we update $\theta$ in the direction of the gradient of the following surrogate objective:

$$\frac{1}{NT} \sum_{k=1}^{N} \sum_{t=0}^{T-1} \left( \log \pi_\theta(a_t^k|h_t^k)\hat{A}_t^k - \lambda \left( R_t^k - V_\theta(h_t^k) \right)^2 - \eta \sum_a \pi_\theta(a|h_t^k) \log \pi_\theta(a|h_t^k) \right) \quad (4)$$

where $\lambda$ and $\eta$ are positive hyperparameters, $\hat{A}_t^k = R_t^k - V_\theta(h_t^k)$, and care is taken not to backpropagate through $\hat{A}_t^k$ as the first term does not provide a useful training signal for $V_\theta$.

## 3 VISUAL ARITHMETIC

We now describe the Visual Arithmetic task domain in detail, as well as the steps required to apply our approach there. We begin by describing the external environment $\mathcal{E}$, before describing the interface $\mathcal{I}$, and conclude the section with a specification of the manner in which $\mathcal{E}$ and $\mathcal{I}$ are coupled in order to produce the POMDP $\mathcal{C}$ with which the controller ultimately interacts.

### 3.1 EXTERNAL ENVIRONMENT

Tasks in the Visual Arithmetic domain can be cast as image classification problems. For each task, each input image consists of an $(n, n)$ grid, where each grid cell is either blank or contains a digit or letter from the Extended MNIST dataset (Cohen et al., 2017). Unless indicated otherwise, we use $n = 2$. The correct label corresponding to an input image is the integer that results from applying a specific (task-varying) reduction operation to the digits present in the image. We consider 5 tasks within this domain, grouped into two kinds. Changing the task may be regarded as changing the external environment $\mathcal{E}$.

**Single Operation Tasks.** In the first kind of task, each input image contains a randomly selected number of digits (2 or 3 unless otherwise stated) placed randomly on the grid, and the agent is required to output an answer that is a function of the both the specific task being performed and the digits displayed in the image. We consider 4 tasks of this kind: Sum, Product, Maximum and Minimum. Example input images are shown in Figure 1, Top Row.

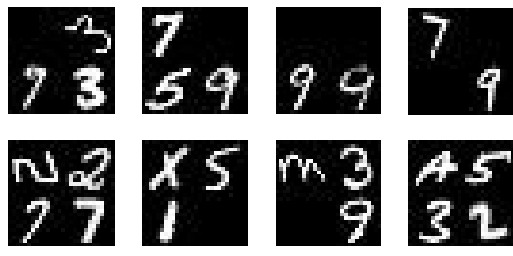

Figure 1: Example input images from the Visual Arithmetic tasks. Top: Input images for any of the 4 Single Operation tasks (Sum, Prod, Max, Min). Correct answer depends on what task they are being used in. For example, when solving the Sum task, correct answers from left to right are: 13, 21, 18, 16. Bottom: Combined task, correct answers from left to right are: 2, 5, 27, 10.

**Combined Task.** We next consider a task that combines the four Single Operation tasks. Each input example now contains a capital EMNIST letter in addition to 2-to-3 digits. This letter indicates which reduction operation should be performed on the digits: $\mathcal{A}$ indicates add/sum, $\mathcal{M}$ indicates multiplication/product, $\mathcal{X}$ indicates maximum, $\mathcal{N}$ indicates minimum. Example input images are shown in Figure 1, Bottom Row. Succeeding on this task requires being able to both carry out all the required arithmetic algorithms *and* being able to identify, for any given input instance, which of the possible algorithms should be executed.

```python
class VisualArithmeticInterface:

    # interface state
    int fovea_x, fovea_y, store, op, digit;
    Image salience_map;

    # pretrained deep nets
    Function op_classifier, digit_classifier, salience_detector;

    def update_interface(ExternalObs e, string action):
        if action == "right": fovea_x += 1
        elif action == "left": fovea_x -= 1
        elif action == "down": fovea_y += 1
        elif action == "up": fovea_y -= 1
        elif action == "+": store += digit
        elif action == "*": store *= digit
        elif action == "max": store = max(store, digit)
        elif action == "min": store = min(store, digit)
        elif action == "+1": store += 1
        elif action == "classify_op":
            op = op_classifier(get_glimpse(e, fovea_x, fovea_y))
        elif action == "classify_digit":
            digit = digit_classifier(get_glimpse(e, fovea_x, fovea_y))
        elif action == "update_salience":
            salience_map = salience_detector(e, fovea_x, fovea_y)
        else:
            raise Exception("Invalid action")
        obs = (fovea_x, fovea_y, store, op, digit, salience_map)
        return obs
```

Figure 2: Pseudo python code (with types added for clarity) for an approximation of the interface used in the Visual Arithmetic task domain.

## 3.2 INTERFACE

We now describe the interface $\mathcal{I}$ that is used to solve tasks in this domain. The first step is to identify information processing functions that we expect to be useful. We can immediately see that for Visual Arithmetic, it will be useful to have modules implementing the following functions:

1. Detect and attend to salient locations in the image.

2. Classify a digit or letter in the attended region.

3. Manipulate symbols to produce an answer.

We select modules to perform each of these functions and then assemble them into an interface which will be controlled by an agent trained via reinforcement learning. A single interface, depicted in Figure 2, is used to solve the various Visual Arithmetic tasks described in the previous section.

This interface includes 3 pre-trained deep neural networks. Two of these are instances of LeNet (LeCun et al., 1998), each consisting of two convolutional/max-pool layers followed by a fully-connected layer with 128 hidden units and RELU non-linearities. One of these LeNets, the *op classifier*, is pre-trained to classify capital letters from the EMNIST dataset. The other LeNet, the *digit classifier*, is pre-trained to classify EMNIST digits. The third network is the *salience detector*, a multilayer perceptron with 3 hidden layers of 100 units each and RELU non-linearities. The salience network is pre-trained to output a salience map when given as input scenes consisting of randomly scattered EMNIST characters (both letters and digits).

### 3.3 $\mathcal{E} - \mathcal{I}$ Coupling and Reinforcement Learning Details

In the Visual Arithmetic setting, $\mathcal{E}$ may be regarded as a degenerate POMDP which emits the same observation, the image containing the EMNIST letters/digits, every time step. $\mathcal{I}$ sends the contents of its *store* field (see Figure 2) to $\mathcal{E}$ every time step as its action. During training, $\mathcal{E}$ responds to this action with a reward that depends on both the time step and whether the action sent to $\mathcal{E}$ corresponds to the correct answer to the arithmetic problem represented by the input image. Specifically, for all but the final time step, a reward of $0$ is provided if the answer is correct, and $-1/T$ otherwise. On the final time step, a reward of $0$ is provided if the answer is correct, and $-1$ otherwise. Each episode runs for $T = 30$ time steps. At test time, no rewards are provided and the contents of the interface's *store* field on the final time step is taken as the agent's guess for the answer to the arithmetic problem posed by the input image. For the controller, we employ a Long Short-Term Memory (LSTM) (Hochreiter and Schmidhuber, 1997) with 128 hidden units. This network accepts observations provided by the interface (see Figure 2) as input, and yields as output both $\pi_\theta(\cdot|h)$ (specifically a softmax distribution) from which an action is sampled, and $V_\theta(h)$ (which is only used during training). The weights of the LSTM are updated according to the actor-critic algorithm discussed in Section 2.3.

## 4 Experiments

In this section, we consider experiments applying our approach to the Visual Arithmetic domain. These experiments involve the high-level tasks described in Section 3.1. For all tasks, our reinforcement learning approach makes use of the interface described in Section 3.2 and the details provided in Section 3.3.

Our experiments look primarily at how performance is influenced by the number of external environment training samples provided. For all sample sizes, training the controller with reinforcement learning requires many thousands of experiences, but all of those experiences operate on the small provided set of input-output training samples from the external environment. In other words, we assume the learner has access to a simulator for the interface, but not one for the external environment. We believe this to be a reasonable assumption given that the interface is designed by the machine learning practitioner and consists of a collection of information processing modules which will, in most cases, take the form of computer programs that can be executed as needed.

We compare our approach against convolutional networks trained using cross-entropy loss, the de facto standard when applying deep learning to image classification tasks. These feedforward networks can be seen as interacting directly with the external environment (omitting the interface) and running for a single time step, i.e. $T = 1$. The particular feedforward architecture we experiment with is the LeNet (LeCun et al., 1998), with either 32, 128 or 512 units in the fully-connected layer. Larger, more complex networks of course exist, but these will likely require much larger amounts of data to train, and here we are primarily concerned with performance at small sample sizes. For training the convolutional networks, we treat all tasks as classification problems with 101 classes. The first 100 classes correspond to the integers 0-99. All integers greater than or equal to 100 (e.g. when multiplying 3 digits) are subsumed under the 101st class.

Experiments showing the sample efficiency of the candidate models on the Single Operation tasks are shown in Figure 3. Similar results for the Combined task are shown in Figure 4. In both cases, our reinforcement learning approach is able to leverage the inductive bias provided by the interface to achieve good performance at significantly smaller sample sizes than the relatively architecturally homogeneous feedforward convolutional networks.

## 5 Related Work

**Production Systems & Cognitive Architectures.** The idea of using a central controller to coordinate the activity of a collection of information processing components has its roots in research on production systems in classical artificial intelligence, and cognitive architectures

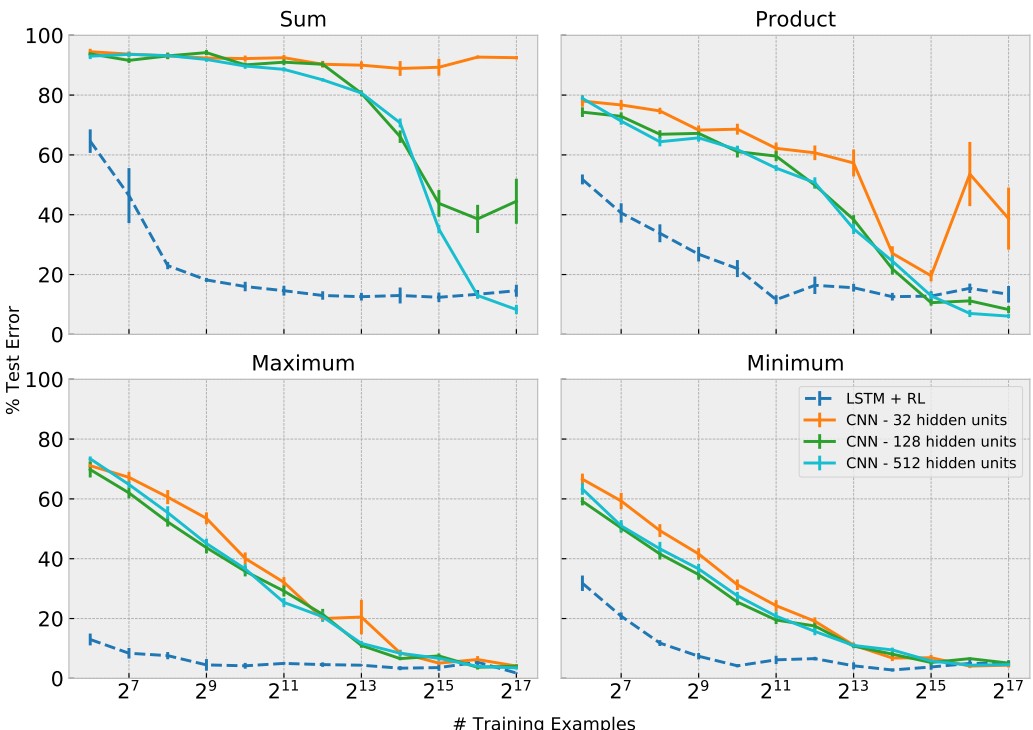

Figure 3: Probing sample efficiency of candidate models on the four Single Operation tasks (Sum, Product, Maximum, Minimum).

in cognitive science. Production systems date back to Newell and Simon's pioneering work on the study of high-level human problem solving, manifested most clearly in the General Problem Solver (GPS; Newell and Simon, 1963). While production systems have fallen out of favour in mainstream AI, they still enjoy a strong following in the cognitive science community, forming the core of nearly every prominent cognitive architecture including ACT-R (Anderson, 2009), SOAR (Newell, 1994; Laird, 2012), and EPIC (Kieras and Meyer, 1997). However, the majority of these systems use hand-coded, symbolic controllers; we are not aware of any work that has applied recent advances in reinforcement learning to learn controllers for these systems for difficult tasks.

**Sequential Models for Supervised Learning.** A related body of work concerns recurrent neural networks applied to supervised learning problems. Mnih et al. (2014), for example, use reinforcement learning to train a recurrent network to control the location of an attentional window in order to classify images containing MNIST digits placed on cluttered backgrounds. Our approach may be regarded as providing a recipe for building similar kinds of models while placing greater emphasis on tasks with a difficult algorithmic components and the use of structured interfaces.

**Neural Abstract Machines.** One obvious point of comparison to the current work is recent research on deep neural networks designed to learn to carry out algorithms on sequences of discrete symbols. Some of these frameworks, including the Differentiable Forth Interpreter (Riedel and Rocktäschel, 2016) and TerpreT (Gaunt et al., 2016), achieve this by explicitly generating code, while others, including the Neural Turing Machine (NTM; Graves et al., 2014), Neural Random-Access Machine (NRAM; Kurach et al., 2015), Neural Programmer (NP; Neelakantan et al., 2015), and Neural Programmer-Interpreter (NPI; Reed and De Freitas, 2015) avoid generating code and generally consist of a controller network that learns to perform actions using a differentiable external computational medium (i.e. a differentiable interface) in order to carry out an algorithm.

Our approach is most similar to the latter category, the main difference being that we have elected not to require the external computational medium to be differentiable, which provides it with greater flexibility in terms of the components that can be included in the interface. In fact, our work is most similar to Zaremba et al. (2016), which also uses reinforcement learning to learn algorithms, and from which we borrowed the idea of an *interface*, the main difference being that we have included deep networks in our interfaces in order to tackle tasks with non-trivial perceptual components.

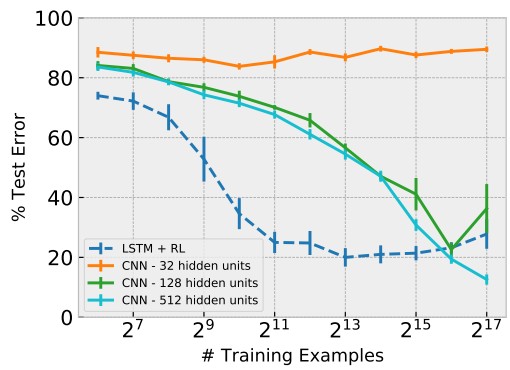

Figure 4: Probing sample efficiency on the Combined task.

**Visual Arithmetic.** Past work has looked at learning arithmetic operations from visual input. Hoshen and Peleg (2016) train a multi-layer perceptron to map from images of two 7-digit numbers to an image of a number that is some task-specific function of the numbers in the input images. Specifically, they look at addition, subtraction, multiplication and Roman-Numeral addition. However, they do not probe the sample efficiency of their method, and the digits are represented using a fixed computer font rather than being hand-written, making the perceptual portion of the task significantly easier.

Gaunt et al. (2017) addresses a task domain that is similar to Visual Arithmetic, and makes use of a differentiable code-generation method built on top of TerpreT. Their work has the advantage that their perceptual modules are learned rather than being pre-trained, but is perhaps less general since it requires all components to be differentiable. Moreover, we do not view our reliance on pre-trained modules as particularly problematic given the wide array of tasks deep networks have been used for. Indeed, we view our approach as a promising way to make further use of any trained neural network, especially as facilities for sharing neural weights mature and enter the mainstream.

**Modular Neural Networks.** Additional work has focused more directly on the use of neural modules and adaptively choosing groups of modules to apply depending on the input. End-to-End Module Networks (Hu et al., 2017) use reinforcement learning to train a recurrent neural network to lay out a feedforward neural network composed of elements of a stored library of neural modules (which are themselves learnable). Our work differs in that rather than having the layout of the network depend solely on the input, the module applied at each stage (i.e. the topology of the network) may depend on past module applications within the same episode, since a decision about which module to apply is made at every time step. Systems built using our framework can, for example, use their modules to gather information about the environment in order to decide which modules to apply at a later time, a feat that is not possible with Module Networks.

In Deep Sequential Neural Networks (DSNN; Denoyer and Gallinari (2014)), each edge of a fixed directed acyclic graph (DAG) is a trainable neural module. Running the network on an input consists in moving through the DAG starting from the root while maintaining a feature vector which is repeatedly transformed by the neural modules associated with the traversed edges. At each node of the DAG, an outgoing edge is stochastically selected for traversal by a learned controller which takes the current features as input. This differs from our work, where each module may be applied many times rather than just once as is the case for the entirely feedforward DSNNs (where no module appears more than once in any path through the DAG connecting the input to the output). Finally, PathNet is a recent advance in the use of modular neural networks applied to transfer learning in RL (Fernando et al., 2017). An important difference from our work is that modules are recruited for the entire duration of a task, rather than on the more fine-grained step-by-step basis used in our approach.

## 6 DISCUSSION AND CONCLUSION

There are number of possible future directions related to the current work, including potential benefits of our approach that were not explored here. These include the ability to take advantage of conditional computation; in principle, only the subset of the interface needed to carry out the chosen action needs to be executed every time step. If the interface contains many large networks or other computationally intensive modules, large speedups can likely be realized along these lines. A related idea is that of adaptive computation time; in the current work, all episodes ran for a fixed number of time steps, but it should be possible to have the controller decide when it has the correct answer and stop computation at that point, saving valuable computational resources. Furthermore, it may be beneficial to train the perceptual modules and controller simultaneously, allowing the modules to adapt to better perform the uses that the controller finds for them. Finally, the ability of reinforcement learning to make use of discrete and non-differentiable modules opens up a wide array of possible interface components; for instance, a discrete knowledge base may serve as a long term memory. Any generally intelligent system will need many individual competencies at its disposal, both perceptual and algorithmic; in this work we have proposed one path by which a system may learn to coordinate such competencies.

We have proposed a novel approach for solving tasks that require both sophisticated perception and symbolic computation. This approach consists in first designing an interface that contains information processing modules such as pre-trained deep neural networks for processing perceptual data and modules for manipulating stored symbolic representations. Reinforcement learning is then used to train a controller to use the interface to solve tasks. Using the Visual Arithmetic task domain as an example, we demonstrated empirically that the interface acts as a source of inductive bias that allows tasks to be solved using a much smaller number of training examples than required by traditional approaches.

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
