# OpenReview forum: "Sequential Coordination of Deep Models for Learning Visual Arithmetic"
_ICLR.cc/2018/Conference — Reject_

### Official Review · AnonReviewer2 · 2017-11-25
**Sequential Coordination of Deep Models for Learning Visual Arithmetic**

**Rating:** 4
**Confidence:** 4

**Review:**

Summary: This work is a variant of previous work (Zaremba et al. 2016) that enables the use of (noisy) operators that invoke pre-trained neural networks and is trained with Actor-Critic. In this regard it lacks a bit of originality. The quality of the experimental evaluation is not great. The clarity of the paper could be improved upon but is otherwise fine. The existence of previous work (Zaremba et al. 2016) renders this work (including its contributions) not very significant. Relations to prior work are missing. But let's wait for the rebuttal phase.

Pros
-It is confirmed that noisy operators (in the form of neural networks) can be used on the visual arithmetic task

Cons
-Not very novel
-Experimental evaluation is wanting

The focus of this paper is on integrating perception and reasoning in a single system. This is done by specifying an interface that consists of a set of discrete operations (some of which involve perception) and memory slots. A parameterized policy that can make use of these these operations is trained via Actor-Critic to solve some reasoning tasks (arithmetics in this case).

The proposed system is a variant of previous work (Zaremba et al. 2016) on the concept of interfaces, and similarly learns a policy that utilizes such an interface to perform reasoning tasks, such as arithmetics. In fact, the only innovation proposed in this paper is to incorporate some actions that invoke a pre-trained neural network to “read” the symbol from an image, as opposed to parsing the symbol directly. However, there is no reason to expect that this would not function in previous work (Zaremba et al. 2016), even when the network is suboptimal (in which case the operator becomes noisy and the policy should adapt accordingly). Another notable difference is that the proposed system is trained with Actor-Critic as opposed to Q-learning, but this is not further elaborated on by the authors.

The proposed system is evaluated on a visual arithmetics task. The input consists of a 2x2 grid of extended MNIST characters. Each location in the grid then corresponds to the 28 x 28 pixel representation of the digit. Actions include shifting the “fovea” to a different entry of the grid, invoking the digit NN or the operator NN which parse the current grid entry, and some symbolic operations that operate on the memory. The fact that the input is divided into a 2x2 grid severely limits the novelty of this approach compared to previous work (Zaremba et al. 2016). Instead it would have been interesting to randomly spawn digits and operators in a 56 x 56 image and maintain 4 coordinates that specify a variable-sized grid that glimpses a part of the image. This would make the task severely more difficult, given fixed pre-trained networks. The addition of the salience network is unclear to me in the context of MNIST digits, since any pixel that is greater than 0 is salient? I presume that the LSTM uses this operator to evaluate whether the current entry contains a digit or an operator. If so, wouldn’t simply returning the glimpse be enough?

In the experiments the proposed system is compared to three CNNs on two different visual arithmetic tasks, one that includes operators as part of the input and one that incorporates operators only in the tasks description. In all cases the proposed method requires fewer samples to achieve the final performance, although given enough samples all of the CNNs will solve the tasks. This is not surprising as this comparison is rather unfair. The proposed system incorporates pre-trained modules, whose training samples are not taken into account. On the other hand the CNNs are trained from scratch and do not start with the capability to recognize digits or operators. Combined with the observation that all CNNs are able to solve the task eventually, there is little insight in the method's performance that can be gained from this comparison.

Although the visual arithmetics on a 2x2 grid is a toy task it would at least be nice to evaluate some of the policies that are learned by the LSTM (as done by Zaremba) to see if some intuition can be recovered from there. Proper evaluation on a more complex environment (or at least on that does not assume discrete grids) is much desired. When increasing the complexity (even if by just increasing the grid size) it would be good to compare to a recurrent method (Pyramid-LSTM, Pixel-RNN) as opposed to a standard CNN as it lacks memory capabilities and is clearly at a disadvantage compared to the LSTM.

Some detailed comments are:

The introduction invokes evidence from neuroscience to argue that the brain is composed of (discrete) modules, without reviewing any of the counter evidence (there may be a lot, given how bold this claim is).

From the introduction it is unclear why the visual arithmetic task is important.

Several statements including the first sentence lack citations.

The contribution section is not giving any credit to Zaremba et al. (2016) whereas this work is at best a variant of that approach.

In the experiment section the role of the saliency detector is unclear.

Experiment details are lacking and should be included.

The related work section could be more focused on the actual contribution being made.

It strikes me as odd that in the discussion the authors propose to make the entire system differentiable, since this goes against the motivation for this work.


Relation to prior work:

p 1: The authors write: "We also borrow the notion of an interface as proposed in Zaremba et al. (2016). An interface is a designed, task-specific machine that mediates the learning agent’s interaction with the external world, providing the agent with a representation (observation and action spaces) which is intended to be more conducive to learning than the raw representations. In this work we formalize an interface as a separate POMDP I with its own state, observation and action spaces."

This interface terminology for POMDPs was actually introduced in:

J.  Schmidhuber. Reinforcement learning in Markovian and non-Markovian environments. In D. S. Lippman, J. E. Moody, and D. S. Touretzky, editors, Advances in Neural Information Processing Systems 3, NIPS'3, pages 500-506. San Mateo, CA: Morgan Kaufmann, 1991.

p 4: authors write: "For the policy πθ, we employ a Long Short-Term Memory (LSTM)"

Do the authors use the (cited) original LSTM of 1997, or do they also use the forget gates (recurrent units with gates) that most people are using now, often called the vanilla LSTM, by Gers et al (2000)?

p 4: authors write: "One obvious point of comparison to the current work is recent research on deep neural networks designed to learn to carry out algorithms on sequences of discrete symbols. Some of these frameworks, including the Differen-tiable Forth Interpreter (Riedel and Rocktäschel, 2016) and TerpreT (Gaunt et al., 2016b), achieve this by explicitly generating code, while others, including the Neural Turing Machine (NTM; Graves et al., 2014), Neural Random-Access Machine (NRAM; Kurach et al., 2015), Neural Programmer (NP; Neelakan- tan et al., 2015), Neural Programmer-Interpreter (NPI; Reed and De Freitas, 2015) and work in Zaremba et al. (2016) on learning algorithms using reinforcement learning, avoid gen- erating code and generally consist of a controller network that learns to perform actions in a (sometimes differentiable) external computational medium in order to carry out an algorithm."

Here the original work should be mentioned, on differentiable neural stack machines:

G.Z. Sun and H.H. Chen and  C.L. Giles and Y.C. Lee and D. Chen. Connectionist Pushdown Automata that Learn Context-Free Grammars. IJCNN-90, Lawrence Erlbaum, Hillsdale, N.J., p 577, 1990.

Mozer, Michael C and Das, Sreerupa. A connectionist symbol manipulator that discovers the structure of context-free languages. Advances in Neural Information Processing Systems (NIPS), p 863-863, 1993.

---

> ### Author Response · Authors · 2017-12-19
> **Response to Detailed Comments**
>
>
> Responding to the detailed comments:
>
> -- The introduction invokes evidence from neuroscience to argue that the brain is composed of (discrete) modules, without reviewing any of the counter evidence (there may be a lot, given how bold this claim is).
>
> We have removed this claim, as we have other sources of motivation that are likely more convincing, less controversial, and tie in better with our experiments; see third paragraph of the introduction in the new version.
>
> -- From the introduction it is unclear why the visual arithmetic task is important.
>
> In the introduction we have provided more motivation for this task choice, with automatic grading of math-exam questions as a potential future application of our approach.
>
> -- Several statements including the first sentence lack citations.
>
> We have added a citation for the first sentence (assuming you meant first sentence of the introduction).
>
> -- The contribution section is not giving any credit to Zaremba et al. (2016) whereas this work is at best a variant of that approach.
>
> We will take this into consideration.
>
> -- In the experiment section the role of the saliency detector is unclear.
> -- Experiment details are lacking and should be included.
>
> The role of the saliency detector is to provide the controller with a low-dimensional representation of the locations of the digits. We have tried using a low-resolution glimpse in place of the saliency detector as suggested, and it does work, but found it to learn more slowly (our hypothesis is that the saliency network is better at providing only information about symbol location, with information about symbol identity stripped away). Additionally, using a network for the saliency detection rather than a simple glimpse provides an additional example of a role for pre-trained networks in our framework, over and above simple classification.
>
> At any rate, we agree that its role may not be clear; we will work on providing more details on the visual arithmetic interface and the experiments in general in an appendix.
>
> -- The related work section could be more focused on the actual contribution being made.
>
> We are not sure which way to go on this, could you be more specific?
>
> -- It strikes me as odd that in the discussion the authors propose to make the entire system differentiable, since this goes against the motivation for this work.
>
> Agreed, removed in the new version.
>
> And finally, with respect to the suggested citations:
>
> 1. The sense in which we and Zaremba et al. use word "interface" is different from the sense in which it is used in the Schmidhuber article. For us, an interface is a designed POMDP that is coupled to the external world and is intended to make learning easier (by providing information processing modules). Zaremba et al. use a similar notion. In the Schmidhuber article, he basically just uses the term "Non-Markovian Interface" to refer to any partially observable environment.
>
> 2. We use the 1997 version of LSTM (default in tensorflow).
>
> 3. References on neural stack machines are very useful and will be incorporated, thanks.

---

> ### Author Response · Authors · 2017-12-19
> **Response to High-Level Comments**
>
> Thank you for the detailed feedback. First, addressing the high-level comments, your major concerns were:
>
> 1. Lack of difficulty.
>
> We agree that it would be somewhat more convincing if the digits were not restricted to a grid. However, the grid-restricted version has precedent in the literature. The paper
>
>    Alexander L Gaunt, Marc Brockschmidt, Nate Kushman, and Daniel Tarlow.  Differentiable    Programs with Neural Libraries. In International Conference on Machine Learning, pages 1213–1222, 2017.
>
> (which is cited in the Related Work section of the new version of the article) tackles a similar domain  the digits are constrained to a grid in all tasks. It therefore seems unfair to count this too heavily against our work.
>
> That said, we will work on applying our approach to a modified version of the task wherein the digits are not restricted to lie on a grid.
>
> 2. Lack of novelty.
>
> While somewhat similar to Zaremba et al., it is our opinion that the addition of perceptual challenges make our tasks different enough from that work to warrant separate exploration. This will become even more true once we are able to make it work without restricting the digits to a grid, as then the application of a pre-trained network to classify a digit becomes less like a noisy version of "reading a digit from a cell" (which is how things are set up in Zaremba et al.), and more like perceiving the world.
>
> Moreover, while the methodology we use may be regarded as a variant of that used by Zaremba et al., both our motivation and experimental manipulations are significantly different. Our motivation is tackling perceptuo-symbolic tasks in a sample efficient way by providing information processing modules, whereas theirs is to see whether algorithms can be learned by reinforcement learning. On the experimental side, they do not compare against feedforward networks as baselines (probably because they are interested in generalizing to longer sequences than were seen during training, which cannot really be handled by feedforward network, though it still might be illuminating to see how a feedforward network would do in their setting if the sequences were kept at a fixed size). Additionally, they make no mention of sample efficiency; they provide their controllers with a training set containing 30,000,000 training characters and do not manipulate this number. So our work is complementary to theirs in that we explore how many unique training examples are required to induce an algorithm using reinforcement learning, which seems to us like a worthwhile endeavour.
>
> 3. Unfair comparison with feedforward networks.
>
> We will work on making the information processing modules available to feedforward networks in order to make the comparison fairer. See response to reviewer #2 for more discussion on this. Additionally, we will look into including baselines with memory as suggested.

---

### Official Review · AnonReviewer1 · 2017-11-29
**A good proposal with limited experimental evidence**

**Rating:** 3
**Confidence:** 4

**Review:**

The paper presents an interesting model to reuse specialized models trained for perceptual tasks in order to solve more complex reasoning tasks. The proposed model is based on reinforcement learning with an agent that interacts with an environment C, which is the combination of E and I, the external world and the interface, respectively. This abstraction is nicely motivated and contextualized with respect to previous work.

However, the paper evaluates the proposed model in artificial tasks with limited reasoning difficulty: the tasks can be solved with simpler baseline models. The paper argues that the advantage of the proposed approach is data efficiency, which seems to be a side effect of having pre-trained modules rather than a clear superior reasoning capability. The paper discusses other advantages of the model, but these are not tested or evaluated either. A more convincing experimental setup would include complex reasoning tasks, and the evaluation of all the aspects mentioned as benefits: computational time, flexibility of computation, better accuracy, etc.

---

> ### Author Response · Authors · 2017-12-19
> **Response**
>
> Thank you for the insightful comments. We have posted an updated version of the article which makes our motivation clearer (see the introduction).
>
> As for your suggestions, we are in the process of running additional experiments which:
>
> 1. Make the comparison with the feedforward network fairer by supplying the feedforward network with the pre-trained modules.
>
> 2. Demonstrate additional benefits of the sequential setup, including ability to adapt the amount of computation to the difficulty of the example, and improved amenability to curriculum learning compared to feed-forward networks.
>
> 3. Tackle more difficult reasoning tasks. It is our hypothesis that our approach will actually do better at more difficult reasoning tasks, and we may be able to find tasks that cannot be learned by feedforward networks at all.

---

### Official Review · AnonReviewer3 · 2017-11-30
**Application of existing classifier networks as components of a visual arithmetic RL problem. OK results, not particularly surprising.**

**Rating:** 2
**Confidence:** 4

**Review:**

Summary: The authors use RL to learn a visual arithmetic task, and are able to do this with a relatively small number of examples, presumably not including the number of examples that were used to pre-train the classifiers that pre-process the images. This appears to be a very straightforward application of existing techniques and networks.

Quality: Given the task that the authors are trying to solve, the approach seems reasonable.
Clarity: The paper appears quite clearly written for the most part.
Originality & Significance: Unless I am missing something important, or misunderstanding something, I do not really understand what is significant about this work, and I don't see it as having originality.

Nitpick 1: top of Page 5, it says "Figure ??"
Nitpick 2: Section 2.3 says "M means take the product, A means take the sum, etc". Why choose exactly those terms that obscure the pattern, and then write "etc"? In Figure 1, "X" could mean multiply, or take the maximum, but by elimination, it means take the maximum. It would have only added a few characters to the paper to specify the notation here, e.g. "Addition(A), Max (X), Min (N), Multiply (M)". If the authors insist on making the reader figure this out by deduction, I recommend they just write "We leave the symbols-operation mapping as a small puzzle for the reader."

The authors might find the paper "Visual Learning of Arithmetic Operations” by  Yedid Goshen and Shmuel Peleg to also be somewhat relevant, although it's different from what they are doing.

Section 3. The story from the figures seems to be that the authors' system works beats a CNN when there are very few examples. But significance of this is not really discussed in any depth other than being mentioned in corresponding places in the text, i.e. it's not really the focal story of the text.

Pros: Seems to work OK. Seems like a reasonable application of pre-trained nets to allow solving a different visual problem for which less data might be available.

Cons: Unless I am missing an important point, the results are unsurprising, and I am not clear what is novel or significant about them.

---

> ### Author Response · Authors · 2017-12-19
> **Response**
>
> Thank you for the insightful comments. We have posted an updated version of the article which fixes the nitpicks (see Section 3.1, paragraph labelled “Combined Task” for our fix to nitpick 2, though the puzzle idea was a close second ;) ) and cites the article that you mentioned. As for the higher-level problems, we address them point-by-point.
>
> 1. Lack of novelty
>
> * Appearance of lack of novelty may stem from incomplete framing/motivation of our approach in the first draft of the paper. In general we are interested in creating systems that can learn to coordinate information processing modules to solve high-level tasks.  The use-case we have in mind involves an ML practitioner who would like to solve a given task, and has at their disposal a collection of information-processing modules that would clearly be useful for the task. The practitioner would like to provide the learning agent with access to those modules (to act as a source of inductive bias, allowing the task to be learned from fewer examples) but does not know how to make those modules available to the learning agent. We propose reinforcement learning as a means of "injecting" the modules into a trainable system; this has the benefit that the modules are not required to be differentiable, greatly expanding the space of modules that may be used. We use the visual arithmetic task as an example of this use-case, wherein it is clear that there are information-processing modules that would be helpful, but it is not clear at first blush how to combine those modules into a system that performs the required task, or that can be trained to perform the task, especially since several of the required operations are symbolic and non-differentiable.
>
>  We have modified the introduction of the article to be clearer about this motivation, which we feel was somewhat lacking in the original draft. If this clarification doesn't alleviate your concerns about novelty, could you expand on these concerns, perhaps citing specific work?
>
> 2. Relating the experiments to the text
>
> The revised introduction also makes it clear that the benefit that we expect to gain from providing the agent with information-processing modules is improved sample efficiency, which is what the experiments show.

---

### Decision · Program_Chairs · 2018-01-29
**ICLR 2018 Conference Acceptance Decision**

**Decision:**

Reject

**Comment:**

The consensus among the reviewers is that this paper is not quite ready for publication for reasons I will summarize in more detail below. However, I think there are some things that are really nice about this approach, and worth calling out:

PROS:

1.  the idea of tackling tasks broadly all the way from perception through symbolic reasoning is an important direction.

2. It certainly would be useful to have a "plug and play" framework in which various knowledge sources or skills can be assembled behind a simple interface designed by the ML practioner to solve a given problem or class of problems.

3. Clearly finding ways to increase sample efficiency -- especially in a deep net approach -- is of great importance practically.

4. The writing  is good.

CONS:

1. The comparison to feedforward networks needs to be made fair in order to disentangle the benefit of the architecture from the benefit of pre-training the modules.

2. Using the very limited 2x2 grid was too low a bar for the reviewers.  The authors aim at a  more general, efficient architecture useful for a variety of tasks, and perhaps you didn't want to devote too much time to this particular task, but I think having a slam-dunk example of the power of the approach is really necessary to be convincing.

3. Given the similarity, I think more has to be done to show the intellectual contribution over Zaremba et al, the difference in motivation notwithstanding.  One way to do this is to really prove out the increased sample efficiency claim.